# COMMUNICATION-EFFICIENT SAMPLING FOR DISTRIBUTED TRAINING OF GRAPH CONVOLUTIONAL NETWORKS

## ABSTRACT

Training Graph Convolutional Networks (GCNs) is expensive as it needs to aggregate data recursively from neighboring nodes. To reduce the computation overhead, previous works have proposed various neighbor sampling methods that estimate the aggregation result based on a small number of sampled neighbors. Although these methods have successfully accelerated the training, they mainly focus on the single-machine setting. As real-world graphs are large, training GCNs in distributed systems is desirable. However, we found that the existing neighbor sampling methods do not work well in a distributed setting. Specifically, a naive implementation may incur a huge amount of communication of feature vectors among different machines. To address this problem, we propose a communication-efficient neighbor sampling method in this work. Our main idea is to assign higher sampling probabilities to the local nodes so that remote nodes are accessed less frequently. We present an algorithm that determines the local sampling probabilities and makes sure our *skewed* neighbor sampling does not affect much to the convergence of the training. Our experiments with node classification benchmarks show that our method significantly reduces the communication overhead for distributed GCN training with little accuracy loss.

## 1 INTRODUCTION

Graph Convolutional Networks (GCNs) are powerful models for learning representations of attributed graphs. They have achieved great success in graph-based learning tasks such as node classification (Kipf & Welling, 2017; Duran & Niepert, 2017), link prediction (Zhang & Chen, 2017; 2018), and graph classification (Ying et al., 2018b; Gilmer et al., 2017).

Despite the success of GCNs, training a deep GCN on large-scale graphs is challenging. To compute the embedding of a node, GCN needs to recursively aggregate the embeddings of the neighboring nodes. The number of nodes needed for computing a single sample can grow exponentially with respect to the number of layers. This has made mini-batch sampling ineffective to achieve efficient training of GCNs. To alleviate the computational burden, various *neighbor sampling* methods have been proposed (Hamilton et al., 2017; Ying et al., 2018a; Chen et al., 2018b; Zou et al., 2019; Li et al., 2018; Chiang et al., 2019; Zeng et al., 2020). The idea is that, instead of aggregating the embeddings of all neighbors, they compute an unbiased estimation of the result based on a sampled subset of neighbors.

Although the existing neighbor sampling methods can effectively reduce the computation overhead of training GCNs, most of them assume a single-machine setting. The existing distributed GCN systems either perform neighbor sampling for each machine/GPU independently (e.g., PinSage (Ying et al., 2018a), AliGraph (Zhu et al., 2019), DGL (Wang et al., 2019)) or perform a distributed neighbor sampling for all machines/GPUs (e.g., AGL (Zhang et al., 2020)). If the sampled neighbors on a machine include nodes stored on other machines, the system needs to transfer the feature vectors of the neighboring nodes across the machines. This incurs a huge communication overhead. None of the existing sampling methods or the distributed GCN systems have taken this communication overhead into consideration.

In this work, we propose a *communication-efficient neighbor sampling* method for distributed training of GCNs. Our main idea is to assign higher sampling probabilities for local nodes so that remote nodes will be accessed less frequently. By discounting the embeddings with the sampling probability, we make sure that the estimation is unbiased. We present an algorithm to generate the sampling probability that ensures the convergence of training. To validate our sampling method, we conduct experiments with node classification benchmarks on different graphs. The experimental results show that our method significantly reduces the communication overhead with little accuracy loss.

## 2 RELATED WORK

The idea of applying convolution operation to the graph domain is first proposed by Bruna et al. (2013). Later, Kipf & Welling (2017) and Defferrard et al. (2016) simplify the convolution computation with localized filters. Most of the recent GCN models (e.g., GAT (Velickovic et al., 2018), GraphSAGE (Hamilton et al., 2017), GIN (Xu et al., 2019)) are based on the GCN in Kipf & Welling (2017) where the information is only from 1-hop neighbors in each layer of the neural network. In Kipf & Welling (2017), the authors only apply their GCN to small graphs and use full batch for training. This has been the major limitation of the original GCN model as full batch training is expensive and infeasible for large graphs. Mini-batch training does not help much since the number of nodes needed for computing a single sample can grow exponentially as the GCN goes deeper. To overcome this limitation, various neighbor sampling methods have been proposed to reduce the computation complexity of GCN training.

**Node-wise Neighbor Sampling:** GraphSAGE (Hamilton et al., 2017) proposes to reduce the receptive field size of each node by sampling a fixed number of its neighbors in the previous layer. PinSAGE (Ying et al., 2018a) adopts this node-wise sampling technique and enhances it by introducing an importance score to each neighbor. It leads to less information loss due to weighted aggregation. VR-GCN (Chen et al., 2018a) further restricts the neighbor sampling size to two and uses the historical activation of the previous layer to reduce variance. Although it achieves comparable convergence to GraphSAGE, VR-GCN incurs additional computation overhead for convolution operations on historical activation which can outweigh the benefit of reduced number of sampled neighbors. The problem with node-wise sampling is that, due to the recursive aggregation, it may still need to gather the information of a large number of nodes to compute the embeddings of a mini-batch.

**Layer-wise Importance Sampling:** To further reduce the sample complexity, FastGCN (Chen et al., 2018b) proposes layer-wise importance sampling. Instead of fixing the number of sampled neighbors for each node, it fixes the number of sampled nodes in each layer. Since the sampling is conduced independently in each layer, it requires a large sample size to guarantee the connectivity between layers. To improve the sample density and reduce the sample size, Huang et al. (2018) and Zou et al. (2019) propose to restrict the sampling space to the neighbors of nodes sampled in the previous layer.

**Subgraph Sampling:** Layer-wise sampling needs to maintain a list of neighbors and calculate a new sampling distribution for each layer. It incurs an overhead that can sometime deny the benefit of sampling, especially for small graphs. GraphSAINT (Zeng et al., 2020) proposes to simplify the sampling procedure by sampling a subgraph and performing full convolution on the subgraph. Similarly, ClusterGCN (Chiang et al., 2019) pre-partitions a graph into small clusters and constructs mini-batches by randomly selecting subsets of clusters during the training.

All of the existing neighbor sampling methods assume a single-machine setting. As we will show in the next section, a straightforward adoption of these methods to a distributed setting can lead to a large communication overhead.

## 3 BACKGROUND AND MOTIVATION

In a $M$-layer GCN, the $l$-th convolution layer is defined as $H^{(l)} = P\sigma(H^{(l-1)})W^{(l)}$ where $H^{(l)}$ represents the embeddings of all nodes at layer $l$ before activation, $H^{(0)} = X$ represents the feature vectors, $\sigma$ is the activation function, $P$ is the normalized Laplacian matrix of the graph, and $W^{(l)}$ is

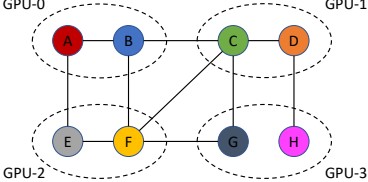 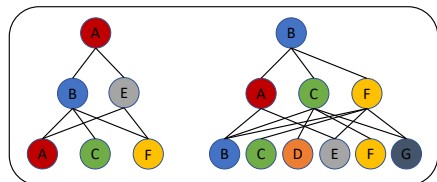

Figure 1: An example of distributed GCN training. Left: A graph with 8 nodes are divided into four parts and stored on four GPUs. Right: For a two-layer GCN, to compute the embedding of node A, we need the feature vectors of node A, B, C, E and F; to compute the embedding of node B, we need the feature vectors of node B, C, D, E, F, G. Nodes that are not on the same GPU need to be transferred through the GPU connections.

the learnable weights at layer $l$. The multiple convolution layers in the GCN can be represented as

$$H^{(M)} = P\sigma(H^{(l-1)}(...\sigma(\underbrace{PXW^{(1)}}_{H^{(1)}})...))W^{(M)}. \tag{1}$$

The output embedding $H^{(M)}$ is given to some loss function $F$ for downstream learning tasks such as node classification or link prediction.

**GCN as Multi-level Stochastic Compositional Optimization:** As pointed out by Cong et al. (2020), training a GCN with neighbor sampling can be considered as a multi-level *stochastic compositional optimization* (SCO) problem (although their description is not accurate). Here, we give a more precise connection between GCN training and multi-level SCO. Since the convergence property of algorithms for multi-level SCO has been extensively studied (Yang et al., 2019; Zhang & Xiao, 2019; Chen et al., 2020), this connection will allow us to study the convergence of GCN training with different neighbor sampling methods. We can define the graph convolution at layer $l \in [1, M]$ as a function $f^{(l)} = P\sigma(H^{(l-1)})W^{(l)}$. The embedding approximation with neighbor sampling can be considered as a stochastic function $f^{(l)}_{\omega_l} = \tilde{P}^{(l)}\sigma(H^{(l-1)})W^{(l)}$ where $\tilde{P}^{(l)}$ is a stochastic matrix with $\mathbb{E}_{\omega_l}[\tilde{P}^{(l)}] = P$. Therefore, we have $f^{(l)} = \mathbb{E}_{\omega_l}[f^{(l)}_{\omega_l}]$. The loss function of the GCN can be written as

$$\mathcal{L}(\theta) = \mathbb{E}_{\omega_{(M+1)}} \left[ f^{(M+1)}_{\omega_{(M+1)}} \left( \mathbb{E}_{\omega_M} \left[ f^{(M)}_{\omega_M} \left( ...E_{\omega_1}[f^{(1)}(\theta)]... \right) \right] \right) \right]. \tag{2}$$

Here, $\theta$ is the set of learnable weights at all layers $\{W^{(1)}, ..., W^{(M)}\}$, $f^{(M+1)} = F(H^{(M)})$, and the stochastic function $f^{(M+1)}_{\omega_{(M+1)}}$ corresponds to the mini-batch sampling.

**Distributed Training of GCN:** As the real-world graphs are large and the compute/memory capacity of a single machine is limited, it is always desirable to perform distributed training of GCNs. A possible scenario would be that we train a GCN on a multi-GPU system. The global memory of a single GPU cannot accommodate the feature vectors of all nodes in the graph. It will be inefficient to store the feature vectors on the CPU main memory and move the feature vectors to GPU in each iteration of the training process because the data movement incurs a large overhead. We want to split the feature vectors and store them on multiple GPUs so that each GPU can perform calculation on its local data. Another possible scenario would be that we have a large graph with rich features which cannot be store on a single machine. For example, the e-commerce graphs considered in AliGraph (Zhu et al., 2019) can 'contain tens of billions of nodes and hundreds of billions of edges with storage cost over 10TB easily'. Such graphs need to be partitioned and stored on different machines in a distributed system. Figure 1 shows an example of training a two-layer GCN on four GPUs. Suppose full neighbor convolution is used and each GPU computes the embeddings of its local nodes. GPU-0 needs to compute the embeddings of node A and B and obtain a stochastic gradient $\tilde{g}_0$ based on the loss function. GPU-1 needs to compute the embeddings of node C and D and obtain a stochastic gradient $\tilde{g}_1$. Similarly, GPU-2 and GPU-3 compute the embeddings of their local nodes and obtain stochastic gradient $\tilde{g}_2$ and $\tilde{g}_3$. The stochastic gradients obtained on different GPUs are then averaged and used to update the model parameters.

**Communication of Feature Vectors:** As shown in Figure 1, the computation of a node's embedding may involve reading the feature vector of a remote node. To compute the embedding of node

A on GPU-0, we need the intermediate embeddings of node B and E, which in turn need the feature vectors of node A, B, C, E and F (Note that the feature vector of node E itself is needed to compute its intermediate embedding; the same for node B). Since node C, E, F are not on GPU-0, we need to send the feature vectors of node C from GPU-1 and node E, F from GPU-2. Similarly, to compute the embedding of node B on GPU-0, we need feature vectors of node B, C, D, E, F and G, which means that GPU-0 needs to obtain data from all of the other three GPUs. This apparently incurs a large communication overhead. Even with neighbor sampling, the communication of the feature vectors among the GPUs are unavoidable. In fact, in our experiments on a four-GPU workstation, the communication can take more than 60% of the total execution time with a naive adoption of neighbor sampling. The problem is expected to be more severe on distributed systems with multiple machines. Therefore, reducing the communication overhead for feature vectors is critical to the performance of distributed training of GCNs.

## 4 COMMUNICATION-EFFICIENT NEIGHBOR SAMPLING

To reduce the communication overhead of feature vectors, a straightforward idea is to skew the probability distribution for neighbor sampling so that local nodes are more likely to be sampled. More specifically, to estimate the aggregated embedding of node $i$'s neighbors (i.e., $\sum_{j \in N(i)} w_{ij} x_j$ where $N(i)$ denotes the neighbors of node $i$, $x_j$ is the embedding of node $j$, and $w_{ij}$ is its weight), we can define a sequence of random variables $\xi_j \sim \text{Bernoulli}(p_j)$ where $p_j$ is the probability that node $j$ in the neighbor list is sampled. We have an unbiased estimate of the result as

$$\sum_{j \in N(i)} \frac{1}{p_j} \xi_j w_{ij} x_j. \tag{3}$$

The expected communication overhead with this sampling strategy is

$$comm\_overhead \propto \mathbb{E}\left[\sum_{j \in R} \xi_j\right] = \sum_{j \in R} p_j \tag{4}$$

where $R$ is the set of remote nodes. Suppose we have a sampling budget $B$ and we denote all the local nodes as $L$. We can let $\sum_{j \in N(i)} p_j = \sum_{j \in L} p_j + \sum_{j \in R} p_j = B$ so that $B$ neighbors are sampled on average. It is apparent that, if we increase the sampling probability of local nodes (i.e., $\sum_{j \in L} p_j$), the expected communication overhead will be reduced. However, the local sampling probability cannot be increased arbitrarily. As an extreme case, if we let $\sum_{j \in L} p_j = B$, only the local nodes will be sampled, but we will not be able to obtain an unbiased estimate of the result, which can lead to poor convergence of the training algorithm. We need a sampling strategy that can reduce the communication overhead while maintaining an unbiased approximation with small variance.

### 4.1 VARIANCE OF EMBEDDING APPROXIMATION

Consider the neighbor sampling at layer $l + 1$. Suppose $S_l$ is the set of sampled nodes at layer $l$. We sample from all the neighbors of nodes in $S_l$ and estimate the result for each of the node using (3). The total estimation variance is

$$V = \mathbb{E}\left[\sum_{i \in S_l} \left\| \sum_{j \in N(S_l)} \frac{1}{p_j} \xi_j w_{ij} x_j - \sum_{j \in N(S_l)} w_{ij} x_j \right\|^2 \right] = \sum_{j \in N(S_l)} \left(\frac{1}{p_j} - 1\right) \|w_{*j}\|^2 \|x_j\|^2. \tag{5}$$

Here $\|w_{*j}\|^2 = \sum_{i \in S_l} w_{ij}^2$ is the sum of squared weights of edges from nodes in $S_l$ to node $j$. Clearly, the smallest variance is achieved when $p_j = 1, \forall j$, and it corresponds to full computation. Since we are given a sampling budget, we want to minimize $V$ under the constraint $\sum_{j \in N(S_l)} p_j \leq B$. The optimization problem is infeasible because the real value of $\|x_j\|^2$ is unknown during the sampling phase. Some prior work uses $\|x_j\|^2$ from the previous iteration of the training loop to obtain an optimal sampling probability distribution (e.g., (Cong et al., 2020)). This however incurs an extra overhead for storing $x_j$ for all the layers. A more commonly used approach is to consider

$\|x_j\|^2$ as bounded by a constant $C$ and minimize the upper bound of $V$ (Chen et al., 2018b; Zou et al., 2019). The problem can be written as a constrained optimization problem:

$$\min \quad \sum_{j \in N(S_l)} \left(\frac{1}{p_j} - 1\right) \|w_{*j}\|^2 C \tag{6}$$

$$\text{subject to} \quad \sum_{j \in N(S_l)} p_j \leq B.$$

$$0 < p_j \leq 1.$$

**The Sampling Method Used in Previous Works:** Although the solution to the above problem can be obtained, it requires expensive computations. For example, Cong et al. (2020) give an algorithm that needs to sort $\|w_{*j}\|^2$ and searches for the solution. As neighbor sampling is performed at each layer of the GCN and in each iteration of the training algorithm, finding the exact solution to (6) may significantly slowdown the training procedure. Chen et al. (2018b) and Zou et al. (2019) adopt a simpler sampling method. They define a discrete probability distribution over all nodes in $N(S_l)$ and assign the probability of returning node $j$ as

$$q_j = \frac{\|w_{*j}\|^2}{\sum_{k \in N(S_l)} \|w_{*k}\|^2}. \tag{7}$$

They run the sampling for $B$ times (without replacement) to obtain $B$ neighbors. We call this sampling method *linear weighted sampling*. Intuitively, if a node is in the neighbor list of many nodes in $S_j$ (i.e., $\|w_{*j}\|$ is large), it has a high probability of being sampled. More precisely, the probability of node $j$ being sampled is

$$p_j = 1 - (1 - q_j)^B \leq q_j B. \tag{8}$$

Plugging (7) into (8) and (6), we can obtain an upper bound of the variance of embedding approximation with this linear weighted sampling method as

$$V_{lnr} = \left(\frac{|N(S_l)|}{B} - 1\right) \sum_{k \in N(S_l)} \|w_{*k}\|^2 C \tag{9}$$

Due to its easy calculation, we adopt this sampling strategy in our work, and we skew the sampling probability distribution to the local nodes so that the communication overhead can be reduced.

## 4.2 Skewed Linear Weighted Sampling

Our idea is to scale the sampling weights of local nodes by a factor $s > 1$. More specifically, we divide the nodes in $N(S_l)$ into the local nodes $L$ and the remote nodes $R$, and we define the sampling probability distribution as

$$q_j = \begin{cases} \frac{s\|w_{*j}\|^2}{\sum_{k \in L} s\|w_{*k}\|^2 + \sum_{k \in R} \|w_{*k}\|^2} & \text{if } j \in L \\ \frac{\|w_{*j}\|^2}{\sum_{k \in L} s\|w_{*k}\|^2 + \sum_{k \in R} \|w_{*k}\|^2} & \text{if } j \in R. \end{cases} \tag{10}$$

Compared with (7), (10) achieves a smaller communication overhead because $\sum_{j \in R} p_j$ is smaller. We call our sampling method *skewed linear weighted sampling*. Clearly, the larger $s$ we use, the more communication we save. Our next task is to find $s$ that can ensure the convergence of the training.

We start by studying the approximation variance with our sampling method. Plugging (10) into (8) and (6), we can obtain an upper bound of the variance as

$$V_{skewed} = \left(\left(\frac{|L|}{sB} + \frac{|R|}{B}\right)\left(\sum_{k \in L} s\|w_{*k}\|^2 + \sum_{k \in R} \|w_{*k}\|^2\right) - \sum_{k \in N(S_l)} \|w_{*k}\|^2\right) C \tag{11}$$

$$= V_{lnr} + \frac{(s-1)|R|}{B} \sum_{k \in L} \|w_{*k}\|^2 C + \frac{(1-s)|L|}{sB} \sum_{k \in R} \|w_{*k}\|^2 C.$$

Note that the variance does not necessarily increase with the skewed neighbor sampling (the last term of (11) is negative).

Since GCN training is equivalent to multi-level SCO as explained in the Background section, we can use the convergence analysis of multi-level SCO to study the convergence of GCN training with our skewed neighbor sampling. Although different algorithms for multi-level SCO achieve different convergence rates (Yang et al., 2019; Zhang & Xiao, 2019; Chen et al., 2020), for general non-convex objective function $\mathcal{L}$, all of these algorithms have the optimality error $(\mathcal{L}(x_{k+1}) - \mathcal{L}^*)$ or $\nabla \mathcal{L}(x_{k+1})$ bounded by some terms that are linear to the upper bound of the approximation variance at each level. This means that if we can make sure $V_{skewed} = \Theta(V_{lnr})$, our skewed neighbor sampling will not affect the convergence of the training. In light of this, we have the following theorem for determining the scaler $s$ in (10).

**Theorem 1.** *When the number of remote nodes $|R| > 0$, with some small constant $D$, if we set*

$$s = \left( \frac{D(|N(S_l)| - B)}{|R|} + \frac{1}{2} \right), \qquad (12)$$

*the training algorithm using our sampling probability in (10) will achieve the same convergence rate as using the linear weighted sampling probability in (7).*

*Proof.* If we set $V_{skewed} \leq D_1 V_{lnr}$ with some constant $D_1$, we can calculate an exact upper bound of $s$ by solving the equations with (7) and (10). The upper bound is $\frac{T_1 + T_2}{2} + \frac{\sqrt{(T_1 + T_2)^2 - 4T_3}}{2}$ where $T_1 = \frac{(D_1 - 1)(|L| + |R| - B)}{|R|} + 1$, $T_2 = \frac{((D_1 - 1)(|L| + |R| - B) + |L|) \sum_{k \in R} \|w_{*k}\|^2}{|R| \sum_{k \in L} \|w_{*k}\|^2}$, $T_3 = \frac{|L| \sum_{k \in R} \|w_{*k}\|^2}{|R| \sum_{k \in L} \|w_{*k}\|^2}$. $|L|$ is the number of local nodes, $|R|$ is the number of remote nodes, and $B$ is the sampling budget. For simple computation, we ignore all the terms dependant on $\sum_{k \in L} \|w_{*k}\|^2$ and $\sum_{k \in R} \|w_{*k}\|^2$, and it gives us a feasible solution $s = \frac{T_1}{2} = \left( \frac{D(|N(S_l)| - B)}{|R|} + \frac{1}{2} \right)$ where $D = \frac{D_1 - 1}{2}$. □

Intuitively, if there are few remote nodes (i.e., $\frac{|N(S_l)|}{|R|}$ is large), we can sample the local nodes more frequently, and (12) gives us a larger $s$. If we have a large sampling budget $B$, the estimation variance of the linear weighted sampling (9) is small. We will need to sample enough remote nodes to keep the variance small, and (12) gives us a smaller $s$.

## 5 EXPERIMENTAL RESULTS

We evaluate our communication-efficient sampling method in this section.

### 5.1 EXPERIMENTAL SETUP

**Platform:** We conducted our experiments on two platforms: a workstation with four Nvidia RTX 2080 Ti GPUs, and eight machines each with an Nvidia P100 GPU in a HPC cluster. The four GPUs in the workstation are connected through PCIe 3.0 x16 slot. The nodes in the HPC cluster are connected with 100Gbps InfiniBand based on a fat-tree topology. Our code is implemented with PyTorch 1.6. We use CUDA-aware MPI for communication among the GPUs. To enable the send/recv primitive in PyTorch distributed library, we compile PyTorch from source with OpenMPI 4.0.5.

**Datasets:** We conduct our experiments on five graphs as listed in Table 1. Cora and CiteSeer are two small graphs that are widely used in previous works for evaluating GCN performance (Zou et al., 2019; Chen et al., 2018b; Zeng et al., 2020). Reddit is a medium-size graph with 233K nodes and an average degree of 492. Amazon is a large graph with more than 1.6M nodes and 132M edges (Zeng et al., 2020). We use the same configurations of training set, validation set, and test set for the graphs as in previous works (Zou et al., 2019; Chen et al., 2018b; Zeng et al., 2020). Youtube is a large graph with more than 1M nodes (Mislove et al., 2007). Each

Table 1: Graph datasets.

| Graph | #Nodes | #Edges |
|---|---|---|
| Cora | 2.7K | 10.5K |
| CiteSeer | 3.3K | 9.2K |
| Reddit | 233K | 58M |
| YouTube | 1.1M | 6.1M |
| Amazon | 1.6M | 132M |

Table 2: Comparison of our sampling method with a naive adoption of the LADIES sampler on Cora and CiteSeer.

| Graph | Sampling Method | F1-Score (%) | Communication Data Size (#nodes) |
|---|---|---|---|
| Cora | `Full` | $74.46 \pm 1.36$ | $402582291 \pm 410933$ |
| | `Our (D=4)` | $74.82 \pm 0.97$ | $316248197 \pm 165161$ |
| | `Our (D=8)` | $75.84 \pm 1.05$ | $299891935 \pm 267364$ |
| | `Our (D=16)` | $75.80 \pm 1.69$ | $284031348 \pm 295241$ |
| | `Our (D=32)` | $74.96 \pm 1.15$ | $270444788 \pm 290565$ |
| CiteSeer | `Full` | $66.54 \pm 1.80$ | $900858804 \pm 595144$ |
| | `Our (D=4)` | $65.58 \pm 2.52$ | $722616380 \pm 588753$ |
| | `Our (D=8)` | $65.50 \pm 2.43$ | $704497231 \pm 518062$ |
| | `Our (D=16)` | $65.36 \pm 2.48$ | $689739665 \pm 438101$ |
| | `Our (D=32)` | $65.64 \pm 2.51$ | $679202038 \pm 413854$ |

node represents a user, and the edges represent the links among users. The graph does not have feature vector or label information given. We generate the labels and feature vectors based on the group information of the nodes. More specifically, we choose the 64 largest groups in the graph as labels. The label of each node is a vector of length 64 with element value of 0 or 1 depending on whether the node belongs to the group. Only the nodes that belong to at least one group are labeled. For feature vector, we randomly select 2048 from the 4096 largest groups. If a node does not belong to any group, its feature vector is 0. We use 90% of the labeled nodes for training and the remaining 10% for testing.

**Benchmark and Settings:** We use a 5-layer standard GCN (as in Formula (1)) to perform node classification tasks on the graphs. For Cora, CiteSeer, Reddit and Amazon whose labels are single value, we use the conventional cross-entropy loss to perform multi-class classification. For YouTube, since the nodes' labels are vectors, we use binary cross entropy loss with a rescaling weight of 50 to perform multi-label classification. The dimension of the hidden state is set to 256 for Cora, CiteSeer and Reddit. The dimension of the hidden state is set to 512 for YouTube and Amazon. For distributed training, we divide the nodes evenly among different GPUs. Each GPU runs sampling-based training independently, with the gradients averaged among GPUs in each iteration.

We incorporate our skewed sampling technique into LADIES (Zou et al., 2019) and Graph-SAINT (Zeng et al., 2020), both of which use linear weighted sampling as in (7). The only difference is that LADIES uses all neighbors of nodes at layer $l$ as $N(S_l)$ when it samples nodes at layer $l+1$, while GraphSAINT considers all training nodes as $N(S_l)$ when it samples the subgraph.

We compare the performance of three versions. The first version (`Full`) uses the original linear weighted sampling and transfers sampled neighbors among GPUs. The second version (`Local`) also uses linear weighted sampling but only aggregates neighboring nodes on the same GPU. The third version (`Our`) uses our skewed sampling and transfers sampled neighbors among GPUs.

## 5.2 RESULTS ON SINGLE-MACHINE WITH MULTIPLE GPUS

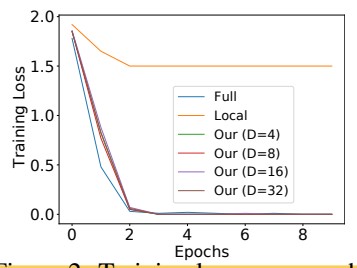

Figure 2: Training loss over epochs on Cora.

We use LADIES to train the GCN with Cora, CiteSeer, Reddit and YouTube graph on the four-GPU workstation. The batch size on each GPU is set to 512, and the number of neighbor samples in each intermediate layer is also set to 512.

**Cora and CiteSeer Results:** Although these two graphs are small and can be easily trained on a single GPU, we apply distributed training to these two graphs and measure the total communication data size to show the benefit of our sampling method. Table 2 shows the best test accuracy and the total communication data size in 10 epochs of training. The results are collected in 10 different runs and we report the mean and deviation of the numbers. Compared with the full-communication version, our sampling method does not cause any accuracy loss for Cora with $D$ (in Formula (12)) set to $4, 8, 16, 32$. For CiteSeer, the mean accuracy in different runs decreases by about 1% with our sampling method. However, the best accuracy in different runs matches the best accuracy of full-communication version. Figure 2

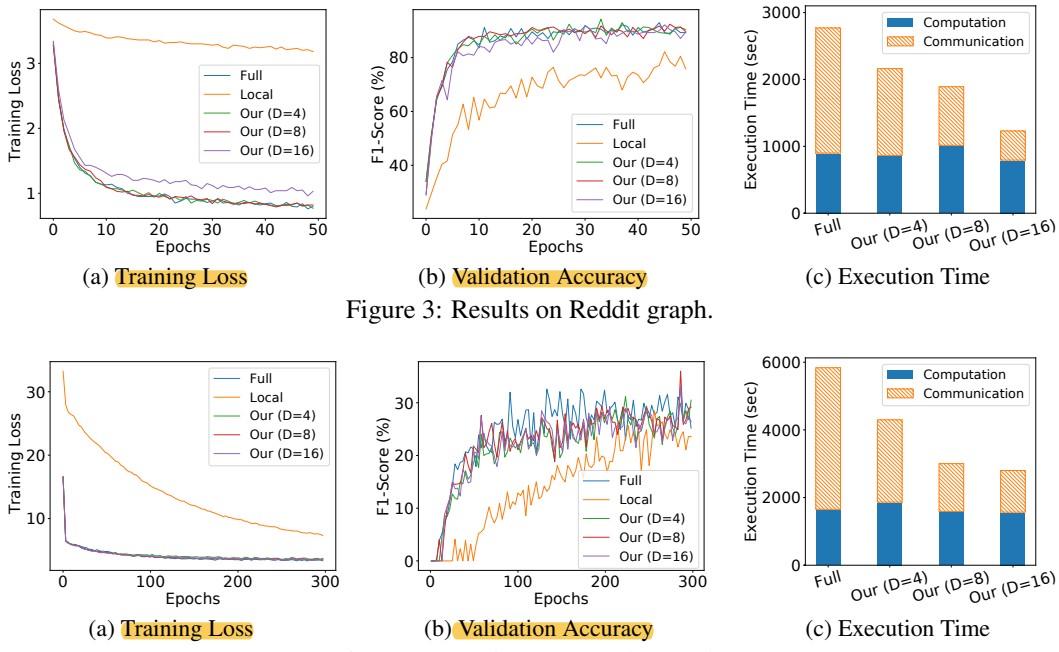

Figure 3: Results on Reddit graph.

Figure 4: Results on YouTube graph.

shows the training loss over epochs with different sampling methods on Cora. We can see that local aggregation leads to poor convergence, due to the loss of information in the edges across GPUs. The other versions have the model converge to optimal after 3 epochs. The training loss on CiteSeer follows a similar pattern. The results indicate that our sampling method does not impair the convergence of training.

The execution times of different versions are almost the same on these two small graphs because the communication overhead is small and reducing the communication has little effect to the overall performance. Therefore, instead of reporting the execution time, we show the communication data size of different versions. The numbers in Table 2 are the numbers of nodes whose feature vectors are transferred among GPUs during the entire training process. We can see that the communication is indeed reduced. When $D = 32$, our sampling method saves the communication overhead by 1.48x on Cora and 1.32x on CiteSeer.

**Reddit and YouTube Results:** These are two larger graphs for which communication overhead is more critical to the performance of distributed training. Figure 3 shows the results on Reddit graph. We run the training for 50 epochs and compare the training loss, validation accuracy and execution time of different versions. The breakdown execution time is shown in Figure 3c. We can see that communication takes more than 60% of the total execution time if we naively adopt the linear weighted sampling method. Our sampling method reduces the communication time by 1.4x, 2.5x and 3.5x with $D$ set to 4, 8, 16, respectively. The actual communication data size is reduced by 2.4x, 3.6x and 5.2x. From Figure 3a, we can see that our sampling method converges at almost the same rate as the full-communication version when $D$ is set to 4 or 8. The training converges slower when $D = 16$, due to the large approximation variance. Figure 3b shows the validation accuracy of different versions. The best accuracy achieved by full-communication version is 93.0%. Our sampling method achieves accuracy of 94.3%, 92.4%, 92.2% with $D$ set to 4, 8, 16, respectively. The figures also show that training with local aggregation leads to a significant accuracy loss.

Figure 4 shows the results on YouTube graph. We run the training for 300 epochs. As shown in Figure 4c, the communication takes more than 70% of the total execution time in the full-communication version. Our sampling method effectively reduces the communication time. The larger $D$ we use, the more communication time we save. The actual communication data size is reduced by 3.3x, 4.6x, 6.7x with $D$ set to 4, 8, 16. Despite the communication reduction, our sampling method achieves almost the same convergence as full-communication version, as shown in Figure 4a and 4b. The full-communication version achieves a best accuracy of 34.0%, while our

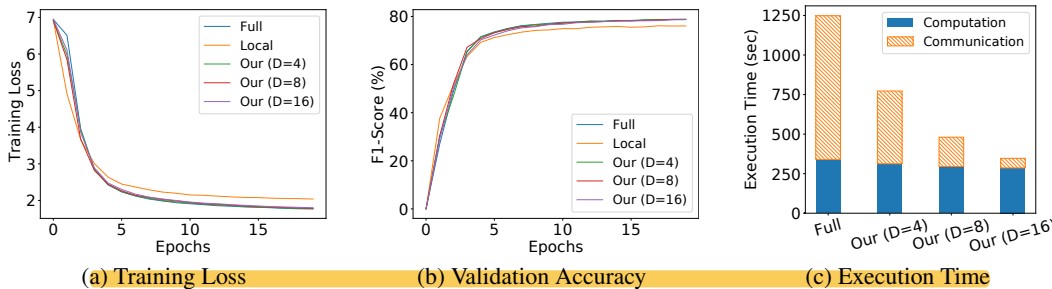

(a) Training Loss        (b) Validation Accuracy        (c) Execution Time

Figure 5: Results on Amazon graph.

sampling method achieves best accuracy of 33.4%, 36.0%, 33.2% with $D$ set to 4, 8, 16. In contrast, local aggregation leads to a noticeable accuracy loss. The best accuracy it achieves is 28.5%.

**Comparison with Centralized Training:** As we described in Section 3, the motivation of partitioning the graph and splitting the nodes' features among GPUs in a single machine is that each GPU cannot hold the feature vectors of the entire graph. As oppose to this distributed approach, an alternative implementation is to store all the features on CPU and copy the feature vectors of the sampled nodes to different GPUs in each iteration. For example, PinSage (Ying et al., 2018a) adopts this approach for training on

Table 3: Execution time of the full-communication version with distributed features on GPUs (Time-Distr) and a centralized version with all features stored on CPU and copied to GPU in each iteration (Time-Centr).

| Graph | Time-Distr (sec) | Time-Centr (sec) |
|---|---|---|
| Cora | 4.8 | 58.1 |
| Citeseer | 4.6 | 59.8 |
| Reddit | 2763.1 | 6622.0 |
| YouTube | 5852.2 | 7028.9 |

large graphs. To justify our distributed storage of the feature vectors, we collect the performance results of GCN training with this centralized storage of feature vectors on CPU. Table 3 lists the training time for different graphs. Even compared with the full-communication baseline in Figure 3c and 4c, this centralized approach is 1.2x to 13x slower. This is because the centralized approach incurs a large data movement overhead for copying data from CPU to GPU. The results suggest that, for training large graphs on a single-machine with multiple GPUs, distributing the feature vectors onto different GPUs is more efficient than storing the graph on CPU.

## 5.3 RESULTS ON MULTIPLE MACHINES

We incorporate our sampling method to GraphSAINT and train a GCN with Amazon graph on eight machines in a HPC cluster. The subgraph size is set to 4500. With our skewed sampling method, a subgraph is more likely to contain nodes on the same machine, and thus incurs less communication among machines. We run the training for 20 epochs. As shown in Figure 5c, the communication takes more than 75% of the total execution time in the full-communication version. Our sampling method effectively reduces the communication overhead. The overall speedup is 1.7x, 2.8x, 4.2x with $D$ set to 4, 8, 16. As shown in Figure 5a and 5b, our sampling method achieves almost the same convergence as full-communication version. The full-communication version achieves a best accuracy of 79.31%, while our sampling method achieves best accuracy of 79.5%, 79.19%, 79.29% with $D$ set to 4, 8, 16. Although the training seems to converge with local aggregation on this dataset, there is a clear gap between the line of local aggregation and the lines of other versions. The best accuracy achieved by local aggregation is 76.12%.

## 6 CONCLUSION

In this work, we study the training of GCNs in a distributed setting. We find that the training performance is bottlenecked by the communication of feature vectors among different machines/GPUs. Based on the observation, we propose the first communication-efficient sampling method for distributed GCN training. The experimental results show that our sampling method effectively reduces the communication overhead while maintaining a good accuracy.

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
