# OpenReview forum: "Communication-Efficient Sampling for Distributed Training of Graph Convolutional Networks"
_ICLR.cc/2021/Conference — Reject_

### Official Review · AnonReviewer1 · 2020-10-27
**a new method for distribtued GNN**

**Rating:** 4
**Confidence:** 5

**Review:**

This paper proposed a new distributed training method for GNNs. Specifically, unlike traditional distributed training methods for CNNs where data points are independent, nodes in a graph are dependent on each other. Thus, this dependence incurs communication between different workers in the distributed training of GNNs. This paper aims to reduce the communication cost in this procedure. Here, this paper proposed to sample more neighbor nodes within the same worker while reducing the sampling probability for the neighbor nodes on other workers. It also provides some theoretical analysis and conducts the experiments to verify the proposed method.

1. The idea is simple. It is just a trad-off between intra-worker sampling and inter-worker sampling. In fact, it does not address the real challenge in distributed training of GNNs. Even though sampling more intra-worker neighbor nodes can reduce the communication cost, it will impair the prediction performance. A good solution should reduce communication costs and try to make the prediction performance as good as possible. However, this method only focuses on the former one.

2. In the proof of Theorem 1, this paper assumes there exists a constant $D_1$, and further claims that $D$ is small. However, no evidence is provided to verify $D$ is small.  Thus, the claim in Theorem 1 does not hold. Moreover, without any knowledge regarding $D$, the bound for $s$ is useless.

3. Regarding experiments, an important baseline is missed. Specifically, the method only using intra-worker neighbor nodes should be used. Otherwise, the current experimental results cannot support the efficacy of the proposed method.

---

> ### Author Response · Authors · 2020-11-18
> **Response to Reviewer1**
>
> Q1:  A good solution should reduce communication costs and try to make the prediction performance as good as possible. However, this method only focuses on the former one.
>
> Our sampling method keeps the estimation variance in the same order as the original sampling method, and thus makes sure the training has the same asymptotic convergence rate. Our experimental results show that our method reduces the communication without impairing the prediction accuracy.
>
> Q2: In the proof of Theorem 1, this paper assumes there exists a constant $D_1$...without any knowledge regarding $D$, the bound for $s$ is useless.
>
> Note that $D$ is a tuning parameter. The point of Formula (12) in Theorem 1 is to show that the scale factor $s$ is dependent on the number of local and remote nodes, instead of being an arbitrary constant value.
>
> Q3: an important baseline is missed. Specifically, the method only using intra-worker neighbor nodes should be used.
>
> This baseline has significant accuracy losses. We will add the result in the revised version.

---

### Official Review · AnonReviewer4 · 2020-10-28
**It is a well-written paper. But the paper lacks novelty. Many alternative methods were not discussed and compared.**

**Rating:** 4
**Confidence:** 5

**Review:**

The paper presents a simple method to reduce the communication for the distributed sampling-based training of GCN. Specifically, communication is reduced by sampling local nodes more and remote nodes less.

The paper is well written, and motivation has been clearly conveyed. Related works in sampling-based training of GCN are well summarized and categorized. The communication ratio of the proposed skewed linear weighted sampling method explored in experiments is around x1 ~ x7 without noticeable performance degradation.

Major concerns:
1) The experiments only explore 4 workers with a compression ratio less than 7. How does the proposed method scale with more workers and larger compression ratios?

2) The proposed sampling method and the original sampling method.

In Section 4.2, the authors managed to achieve the same sampling variance for linear weighted sampling and the proposed skewed linear weighted sampling, which may partially due to that the variance of the original linear weighted sampling is not optimal. How will the idea of sampling local nodes more frequently work for other sampling methods?

In experiments, the authors compared the proposed skewed linear weighted sampling with the LADIES sampling method. Why is the original linear weighted sampling method not compared with the proposed skewed linear weighted sampling?

3) It seems that the authors distribute the graph to workers and each worker only holds the local nodes' original data and intermediate activations during training. How the graph is distributed in experiments needs more clarification.

4) The importance of distributed sampling-based training of GCN is not explained. Naive distributed training as in 3) is certainly inefficient involving too much communication of neighborhood nodes' features. However, there seem to be some simple alternatives of 3) as in 4.1) and 4.2).

4.1) Sampling-based training of GCN already makes it feasible to train GCN on large datasets in the single-machine setting. Why can't we just use multiple workers to do the training as in single-machine settings and then average the gradient or average the model periodically like Local SGD [1]?

4.2) Cluster-GCN may also be another alternative as it also makes it feasible to train large GCN in the single-machine setting. Can we make it work for distributed training the same way as in 4.1)?

References:
[1] Stich, Sebastian U. "Local SGD converges fast and communicates little." arXiv preprint arXiv:1805.09767 (2018).
[2] Chiang, Wei-Lin, et al. "Cluster-GCN: An efficient algorithm for training deep and large graph convolutional networks." Proceedings of the 25th ACM SIGKDD International Conference on Knowledge Discovery & Data Mining. 2019.

---

> ### Author Response · Authors · 2020-11-18
> **Response to Reviewer4**
>
> Q1: The experiments only explore 4 workers with a compression ratio less than 7. How does the proposed method scale with more workers and larger compression ratios?
>
> Although our experiments focus on multiple GPUs on a single machine, our method is applicable to distributed systems with multiple machines. We tested our method on 8 machines and the performance follows a similar pattern. The speedup is even higher since communication is more expensive. We will add the result in the updated version.
>
> Q2: How will the idea work for other sampling methods? The authors compared the proposed method with the LADIES sampling method. Why is the original linear weighted sampling method not compared with the proposed skewed linear weighted sampling?
>
> Note that LADIES, FastGCN and GraphSAINT all use linear weighted sampling. The only difference is the definition of $N(S_l)$ in Formula (7). Therefore, our method is applicable to all these sampling methods. We incorporated our method into GraphSAINT and tested with the graph used in GraphSAINT paper. The performance and accuracy follow a similar pattern as LADIES. We will add the result in the updated version.
>
> Q3: How the graph is distributed in experiments needs more clarification.
>
> The distribution of the graph is shown in Figure1 and is described in Section3 "Distributed Training of GCN".
>
> Q4: Naive distributed training is certainly inefficient involving too much communication of neighborhood nodes' features.
>
> As we described in Section3 "Distributed Training of GCN", we target multi-GPU platforms where each GPU is not large enough to hold the entire graph. We also target multi-machine platforms where the graph is too large to be duplicated on each machine.  The alternatives described in 4.1) and 4.2) do not work for these cases.
>
> Q5: the paper lacks nolvety...
>
> To the best of our knowledge, our paper is the first to consider reducing the communication overhead of nodes' features in the setting described in Q4.

---

### Official Review · AnonReviewer3 · 2020-10-29
**A simple approach to speeding up training of GCNs in distributed systems (but it works)**

**Rating:** 6
**Confidence:** 3

**Review:**

The paper presents a sampling-based approach to speeding-up training of GCNs in distributed systems. The key step in this task involves exchanging and aggregating messages sent along the edges of the graph. If the nodes of the graph are partitioned between several machines, then exchanging those messages involve costly communication between the nodes. To reduce this communication, the paper introduces a variant of the node sampling approach of Chen et al. (2018b) and Zou et al. (2019), where the probabilities of nodes in other machines are scaled down by some factor s. The approach is evaluated experimentally, showing that it reduces the amount of communication while (essentially) preserving the accuracy.

PROS:
- The first implementation of sampling-based training for GCNs in the distributed scenario (this is according to the authors, but I am not aware of any prior work of this type either)
 - Experimental evaluation shows significant improvement to the communication cost

CONS:
- The new sampling process is a relatively simple modification of prior work, so there isn't much conceptual novelty in here

Overall, a solid contribution to the area.

---

### Official Review · AnonReviewer5 · 2020-11-04
**Importance weighted sampling for GNN distributed training**

**Rating:** 5
**Confidence:** 3

**Review:**

This work considers the challenge of distributed training for GNNs. The approach is a locality-aware importance weighted sampling procedure. I was not given much time to read the paper but it seems like a decent contribution, albeit too minor of a contribution to the existing literature to be considered a bonafide research paper.

### Quality

- There is nothing clearly wrong with the paper. I did not have time to go through all the equations but I can believe the approach.

### Clarity

- The writing is clear and the approach is well described.

### Originality

There is not a whole lot of originality. Prior work (appropriately cited) has consider a similar approach. The main difference is the locality of sampling to avoid communication.

### Significance of this work

- Important topic, not exciting as a research paper.

### Pros

- Scalability of GNNs is a very important topic that deserves more attention.

- The writing is good; the reader can quickly understand the approach and the main points of the paper. It also helps that the approach is well-known and relatively simple.

### Cons

- Experiments: For a work studying distributed training over graphs with "billions of nodes", it is certainly disappointing to see that the datasets contain up to 1.1M nodes.

- The work seems like a direct application of importance weighting and stratified sampling to sampling the neighborhood of node in a GNN. Locality-aware importance sampling is a common approach used in industry, and rather trivial as a method.


### Other comments

- Regarding the bound V, it would be nice to get a sense of its magnitude. It does not look very efficient. What if we performed a push sampling operation (where the node that has x_j will sample it with probability ||x_j||^2  and push it to the servers that need it) rather than the proposed pull sampling (where each node requests the samples)? That way we don't need to guess or bound the value of ||x_j||^2. Just a quick thought.

-------

The rebuttal did not meaningfully addressed my concerns.

Apologies to the authors for not providing a reference for my comment on approaches for reducing communication in graph-optimization methods being widely known in industry. GraphLab is an example (https://en.wikipedia.org/wiki/GraphLab)

- Y. Low, J. Gonzalez, A. Kyrola, D. Bickson, C. Guestrin and J. Hellerstein. GraphLab: A New Framework for Parallel Machine Learning. In the 26th Conference on Uncertainty in Artificial Intelligence (UAI), Catalina Island, USA, 2010
- Yucheng Low, Joseph Gonzalez, Aapo Kyrola, Danny Bickson, Carlos Guestrin and Joseph M. Hellerstein (2012). "Distributed GraphLab: A Framework for Machine Learning and Data Mining in the Cloud." Proceedings of Very Large Data Bases (PVLDB).
- Joseph Gonzalez, Yucheng Low, Haijie Gu, Danny Bickson, Carlos Guestrin (2012). "PowerGraph: Distributed Graph-Parallel Computation on Natural Graphs." Proceedings of Operating Systems Design and Implementation (OSDI).

---

> ### Author Response · Authors · 2020-11-23
> **Response to Reviewer5**
>
> Q1: it is certainly disappointing to see that the datasets contain up to 1.1M nodes.
>
> Our experiments focus on a single-machine multi-GPU platform. On such a platform, this graph is large enough since each GPU cannot hold the feature vectors and intermediate data of the entire graph.  We also added a new set of results with a graph with 1.6M nodes and 132M edges.
>
> Q2: Locality-aware importance sampling is a common approach used in industry, and rather trivial as a method.
>
> We are not aware of previous works using this method for reducing communication in GCN training.

---

### Author Response · Authors · 2020-11-23
**Added more experimental results**

We appreciate the valuable feedback from all the reviewers. To address the concerns and improve the clarity, we made the following changes to the paper. All the changes are marked in yellow in the revised version.


1. To address the concern of scalability of our method from Reviewer4 and 5, we added a new set of results on eight machines. (Section 5.3)


2. To address the concern of applicability of our method to other sampling methods from Reviewer4, we incorporated our skewed sampling method into the subgraph sampling procedure in GraphSAINT. The new set of results is collected with GraphSAINT.  (Section 5.3)


3. To clarify the setting of our experiments and address the concern of novelty from Reviewer4, we conducted an experiment with centralized storage of the graph on CPU and show that splitting nodes' features among GPU is more efficient. ('Comparison with Centralized Training' in Section 5.2)


4. To address the concern of missing baseline from Reviewer1, we added the training loss and accuracy with only local neighbor aggregation in Figure 2,3,4,5. The results show that this baseline has noticeable accuracy losses.

---

### Decision · Program_Chairs · 2021-01-07
**Final Decision**

**Decision:**

Reject

**Comment:**

The paper introduces a new locality-aware importance weighted sampling procedure for distributed training of GNNs. While the paper is interesting, the reviewers raised some fundamental concerns about it.

The focus on the paper is on scalable methods and the experiments or only run on medium-size datasets(<2m nodes). For such a paper larger scale experiments are expected.

Furthermore, the novelty of the paper is limited.

Overall, the paper is below the high acceptance bar of ICLR.